# A Census and Categorization Method of Epitranscriptomic Marks

**DOI:** 10.3390/ijms21134684

**Published:** 2020-06-30

**Authors:** Julia Mathlin, Loredana Le Pera, Teresa Colombo

**Affiliations:** 1Department of Life Sciences and Medicine, University of Luxembourg, L-4367 Belvaux, Luxembourg; 2CNR-Institute of Biomembranes, Bioenergetics and Molecular Biotechnologies (IBIOM), 70126 Bari, Italy; 3CNR-Institute of Molecular Biology and Pathology (IBPM), 00185 Rome, Italy; teresa.colombo@cnr.it

**Keywords:** epitranscriptome, RNA modifications, epigenetic regulation, gene-expression regulation, post-transcriptional regulation, epitranscriptomics

## Abstract

In the past few years, thorough investigation of chemical modifications operated in the cells on ribonucleic acid (RNA) molecules is gaining momentum. This new field of research has been dubbed “epitranscriptomics”, in analogy to best-known epigenomics, to stress the potential of ensembles of RNA modifications to constitute a post-transcriptional regulatory layer of gene expression orchestrated by writer, reader, and eraser RNA-binding proteins (RBPs). In fact, epitranscriptomics aims at identifying and characterizing all functionally relevant changes involving both non-substitutional chemical modifications and editing events made to the transcriptome. Indeed, several types of RNA modifications that impact gene expression have been reported so far in different species of cellular RNAs, including ribosomal RNAs, transfer RNAs, small nuclear RNAs, messenger RNAs, and long non-coding RNAs. Supporting functional relevance of this largely unknown regulatory mechanism, several human diseases have been associated directly to RNA modifications or to RBPs that may play as effectors of epitranscriptomic marks. However, an exhaustive epitranscriptome’s characterization, aimed to systematically classify all RNA modifications and clarify rules, actors, and outcomes of this promising regulatory code, is currently not available, mainly hampered by lack of suitable detecting technologies. This is an unfortunate limitation that, thanks to an unprecedented pace of technological advancements especially in the sequencing technology field, is likely to be overcome soon. Here, we review the current knowledge on epitranscriptomic marks and propose a categorization method based on the reference ribonucleotide and its rounds of modifications (“stages”) until reaching the given modified form. We believe that this classification scheme can be useful to coherently organize the expanding number of discovered RNA modifications.

## 1. Introduction

Pseudouridine (C5-glycoside isomer of uridine), the first post-transcriptional ribonucleic acid (RNA) modification identified, was discovered in 1957 [1], but, only decades later, the term “epitranscriptomics” was created to describe how that and many other modifications are not genetically encoded but, rather, added on top—thus the prefix “epi” taken from the Greek that means “on top of”—of the transcribed nucleotides [2]. Interest in the field refreshed in the early 2010s, stimulated by evidence that chemical modifications of RNA can be reversible [3] and by technical progress that revealed a huge array of such epitranscriptomic marks decorating RNA molecules [4,5]. Chemical modifications can affect all four ribonucleosides—adenosine (A), guanosine (G), cytidine (C), and uridine (U)—and may imply either covalent addition of a chemical group (such as a methyl or thiol group) or editing of the original RNA sequence, such as in the adenosine-to-inosine (A-to-I) replacement. About two-thirds of epitranscriptomic marks involve the addition of methyl groups. Among others, 1-methyladenosine (m1A), 3-methylcytidine (m3C), and 7-methylguanosine (m7G) are the most common RNA modifications implying base methylation and cause gain of a positive electrostatic charge [6]. The corresponding electro-chemical alteration resulting from the positive charge of m1A, in particular, was one of the first deeply investigated in transfer RNA (tRNA) structure-function relationship [7]. Figure 1 illustrates the chemical formulas for the main types of RNA modifications, including examples of methylation processes, such as for the 6-methyladenosine (m6A), 5-methylcytidine (m5C), and N2-N2-dimethylguanosine (m2,2G), as well as other chemical modifications, such as thiolation of U to form 2-thiouridine (s2U) or its isomerization leading to pseudouridine (Ψ), or editing events, such as converting A to inosine (I). Some modifications can be the result of complex enzymatic mechanisms, requiring multiple steps and involving diverse catalytic proteins and cofactors (e.g., peroxywybutosine or o2yW), while others can be obtained through simple changes (e.g., m1A).

About 160 distinct RNA modifications have been identified so far [8], spread across kingdoms, species, and molecule types. In Appendix A, a comprehensive collection of them is provided, along with updated information, whenever available, about: the natural ribonucleoside from which they derive, the kingdom (eukarya, bacteria, or archaea) in which they have been observed, their target RNA type, their location or preferred motif of occurrence in the host RNA molecule for human and/or other species, and the cognate RNA-modifying enzymes. Importantly, each RNA modification yields specific properties that can play roles well beyond simple structural fine-tuning and crucial impact on structure, folding, and stability, as well as cellular localization and function [9,10,11,12].

In particular, leveraging on the analogy with epigenomics, epitranscriptomic marks may offer a prevalent layer of gene-expression regulation by constituting a substrate for differential binding with RNA-interacting partners performing as effectors of this code. In fact, the epigenetic paradigm of specific sets of regulatory marks made operational by an ensemble of writers, readers, and erasers can also be applied to epitranscriptomics [13,14]. The most abundant internal modification in polyadenylated RNAs of eukaryotes, that is m6A, provides the best example of that paradigm, with methyltransferases (“writers”), binding proteins (“readers”), and demethylases (“erasers”) having been elucidated, along with their interplay and functional outcome [14,15]. Indeed, it is reasonable to expect many more examples to follow. For instance, the human genome includes over 1500 genes encoding for RNA-binding proteins (RBPs) interacting with all known classes of RNA molecules, but the function of at least one-third of them is still unknown [16]. Moreover, many of them (∼200) are annotated in the Online Mendelian Inheritance in Man (OMIM) [17] database as associated with human diseases, suggesting their involvement in important cell processes.

The case of m6A is also useful to illustrate the close link between the field of epitranscriptomics and the high-throughput sequencing (HTS) technologies. In fact, in addition to traditional experimental protocols for RNA modification detection, including thin-layer chromatography (TLC), high-performance liquid chromatography (HPLC), and liquid chromatography-mass spectrometry (LC-MS(MS)) [18], availability of adequate experimental protocols for transcriptome-wide mapping of modification sites and monitoring of their cell dynamics is a key prerequisite to systematic investigation of biological functions. As detailed above, transcripts can either be edited (with base replacement) or covalently linked to small molecules. While base changes can be detected directly by using next-generation sequencing (NGS) technologies, such as RNA-sequencing (RNA-seq), this conventional HTS approach erases information about other types of modification that do not introduce changes in base pairing. Specifically, loss of information occurs during the reverse transcription step, mandatory in standard NGS protocols, that converts RNA into complementary DNA (cDNA) before amplification and sequencing. Consequently, modifications that do not affect Watson–Crick base pairing during cDNA synthesis require dedicated assays based on, for example, immunoprecipitation with ad hoc antibodies. In 2012, by exploiting the advantages of two established technologies, such as immunoprecipitation followed by HTS, two independent groups [4,5], with little technical difference between their experimental strategies, were able to map the transcriptome-wide localization of m6A at high resolution, identifying ∼10,000 m6A sites in the mammalian transcriptomes, clustered near stop-codons and in 3’ untranslated regions (3’UTRs). They used a highly m6A-specific antibody to immunocapture only methylated RNA fragments (after whole transcriptome random fragmentation), which were then sequenced by HTS. Sites of occurrence of m6A modification were localized as “peaks” of signal enrichment relative to input control. Unfortunately, the m6A case is currently more an exception than a rule in this regard. In fact, while the same approach could in principle be extended to other RNA modifications, as well, specific antibodies are currently available only for a small subset of modified ribonucleotides. Thus far, immunoprecipitation-based approaches have been successfully applied to map m1A [19], m5C [20], 5-hydroxymethylcytidine (hm5C) [21], and N4-acetylcytidine (ac4C) [22].

Other methods exploit the natural consequence of a handful of RNA modifications to induce the reverse transcriptase (RT) to arrest or to incorporate non-complementary nucleotides into the nascent cDNA. This approach has been used, for example, to predict modifications, such as m1A, m3C, 1-methylguanosine (m1G), I, 1-methylinosine (m1I), and m2,2G, from their so-called RT-signature in human tRNAs [23,24,25].

Finally, other protocols try to make RNA modifications similarly detectable in the cDNA sequence by pre-treatment with chemical compounds producing conformational changes that induce mismatches with the reference sequence or RT arrest, with an accumulation of sequencing reads with identical ends. For example, experimental protocols for selective chemical modification and identification of modified residues are available for dihydrouridine (D), I, m5C, m7G, 2’-O-methylation (Nm), and Ψ [25].

Although these detection methods have provided valuable information in the last years, they can be used to investigate only a minority of epitranscriptomic marks because of the limited availability of antibodies (likely due to the small size of the antigen, the modified ribonucleoside) and the lack of chemical compounds selectively reactive towards a particular RNA modification (an updated overview of the sequencing methods for RNA modification mapping is provided in Reference [25]). Moreover, these methods often require complex and time-consuming protocols and still have some limitations (mainly caused by RNA fragmentation) with respect to specific isoform detection, information about strand-specificity, and occurrence of multiple methylation sites along the same transcript [26]. Furthermore, abundant post-transcriptional modifications can bias detection and quantification of both transcripts and epitranscriptomic marks by interference with cDNA synthesis [27,28]. To circumvent these problems, third-generation sequencing technologies, specifically the Pacific BioSciences (PacBio) [29] and the Oxford Nanopore Technologies (ONT) [30], have been proposed as a new opportunity to detect epitranscriptomic marks more efficiently [31]. PacBio performs single-molecule real-time (SMRT) isoform sequencing by sequencing full-length transcripts with a mean read length of roughly 10 kilobases. This technology uses reverse transcriptase, which incorporates modified bases more slowly than it does with unmodified ones. Therefore, RNA modifications can be distinguished as having specific “kinetic signatures” [32]. The Nanopore sequencer, in turn, can perform single-molecule long sequencing directly on native RNA through a nanopore embedded in a membrane. This Nanopore sequencer can measure disruptions in the current intensity, also known as “squiggles”, compared to raw current intensities, as the RNA or DNA molecule passes through the pore. This technology is able, in principle, to identify the corresponding transiting nucleotides. In particular, Nanopore has been applied to a few DNA and RNA modifications, such as m5C and m6A in DNA, as well as m6A in RNA [25,31,33]. In comparison with previous NGS technologies, PacBio and ONT show the great advantage of increased read length and single nucleotide resolution, but technical limitations still remain, partially due to a faster RNA degradation compared to DNA, and its tendency to fold in loops and knots, making sequencing more difficult. Finally, PacBio sequencing output has currently a significantly higher error rate (10–15%) compared to NGS (<2%), and ONT yields an output with error rates even higher (accuracy between 65–88%) [34]. Overall, we currently lack standard methods for detecting epitranscriptomic marks using next- and third-generation sequencing and rather specialized protocols has to be developed for each case of interest. This issue is mirrored by a similar lack of well-established bioinformatics protocols for identification and annotation of diverse RNA modifications, as well as lack of accurate statistical approaches to cope, in particular, with false positives that can arise at numerous levels in data analysis [35,36]. In fact, one of the first HTS-based systematic investigation of RNA editing had reported all 12 different types of possible base changes, corresponding to nucleotide mismatches of sequencing reads with respect to the reference genome [37], but it was subsequently discovered that all but the A-to-I editing sites were actually false positive calls [38]. m6A stands out as a special case in this regard, because an advanced algorithm, based on machine-learning approaches, has been recently developed and trained to identify m6A transcriptome marks from RNA reads generated by ONT, with results achieving an overall accuracy of about 90% [39]. Importantly, conservation can be used to reduce the chance of false positives in HTS studies [40] since locations of RNA modification tend to be more conserved across species [41] and among individuals than most single nucleotide polymorphisms (SNPs) [42]. In conclusion, only a dozen RNA modifications out of the many reported have sequencing methods available for transcriptome-wide assaying, to date. Furthermore, most of the existing techniques for the detection and validation of RNA modifications still need improvements to increase their efficacy, reduce the number of false positives, and lower the impact of binding biases in antibody-based co-immunoprecipitation methods [43,44]. However, the tremendous boost in innovative sequencing technologies that we are witnessing today holds the promise to fill this gap soon and to provide the epitranscriptomics field with suitable technologies for accurate, comprehensive, and combinatorial investigation of RNA modification marks.

Here, we present a census of currently known RNA modifications, along with a proposed method for classifying these epitranscriptomic marks based on the natural ribonucleotide subject of the modification process and the number of steps occurred until reaching the final output.

## 2. A Categorization Method for RNA Modifications

Even though the number of discovered epitranscriptomic marks is continuously growing, a standard method to represent this relevant list has not been found yet. However, a systematic and organic way to classify already known and newly discovered RNA modifications can be helpful to cope with the complexity that characterizes this field. In fact, epitranscriptomics already entails an order of magnitude higher number of annotated marks compared to epigenomics (∼160 vs. 17).

Here, we propose a categorization method based on the subsequent modification stages leading from the natural RNA base to its final form. This categorization aims to place all known events in a coherent scheme, readily providing the chemical context and the step-by-step process by which the modification occurred (Figure 2a and Appendix A). According to this method, post-transcriptional RNA modifications can be described as a tree that has branches in up to nine different stages, as well illustrated by the case of guanosine-derived chemical modifications (Figure 2b and Figure 3, and Appendix A). The proposed scheme currently accounts for a total of 134 known RNA modifications from archaea, bacteria, and eukarya and can be easily extended to include newly discovered epitranscriptomic marks. A few known RNA modifications are currently not represented in the proposed tree-representation. Of them, three lack minimal amount of information on their biosynthesis to allow proper placement in the scheme (listed in Figure 2a, bottom-right inset), while 21 additional ones are known to modify nascent trascripts at the 5${}^{\prime }$- or 3${}^{\prime }$-end (listed as “nascent transcript modifications” in Appendix A) but cannot be rooted to a specific ribonucleoside and are not discussed in this review.

Starting from the original ribonucleosides (A, C, G, or U), first-stage RNA modifications originate by means of a chemical modification made to the unmodified base. In turn, second-stage modifications originate from a further step of chemical modification following the first one, and so on for any further stage of modification in the proposed categorization scheme, where any featured product derives from chemical modification of a substrate that is the output of a modification occurred at the previous-stage. In this scheme, a shift from the previous state (stage N) to the next one (stage N+1) represents a chemical modification catalyzed by one or more enzymes and leading to a functional modified ribonucloside that has been reported in some RNA species. The number corresponding to each stage is equal to the total number of modifications added to the initial base; thus, it is related to the chemical reaction complexity required to reach that specific stage.

Apart from providing a useful categorical frame to organize growing knowledge about epitranscriptomic marks, our method highlights some inspiring features about the distribution of RNA modifications with respect to original RNA bases. For instance, the number of first-stage modifications in the scheme appears to be similar among all branches, ranging between 7 and 13 (Figure 2a). Conversely, the next level shows an uneven distribution among the four branches, with the uridine’s branch accounting alone for nearly half of the total number of second-stage modifications (24/54) (Figure 2a). Moreover, the overall total of RNA modifications derived from uridine (N = 55) far exceeds those derived from any other RNA base (N = 32, 18, and 29 for adenosine-, cytidine-, and guanosine-derived modifications, respectively) (Figure 2a). Of note, the outstanding number of uridine-derived modifications, especially compared to the much smaller number of cytidine-derived modifications, could mask a contribution actually derived from cytidines. The rationale for this hypothesis is that cytidines can be enzymatically converted to uridines [45,46]. In addition, some cytidine-derived bases, such as the m3C, can be directly converted to 3-methylcytidine (m3U) [47]. A global tree-representation, including modified derivatives for all four ribonucleosides and clearly showing these cross-talks between cytidine- and uridine-derived chemical modifications, is available in Appendix A. Concerning height of the tree, guanosine shows up in the scheme as the only tree featuring up to nine stages of modification due to the complex biosynthetic pathway leading to formation of the o2yW derivative and the other known epitranscriptomic modifications generated along that path (Figure 2b).

Finally, the uridine- and adenosine-derived trees show more leafy branches with multiple derivatives for several nodes (Figure 3a,c), different from the tree of cytidine-derived modifications, where most nodes have none or just one derivative (Figure 3b).

In the next sections, best characterized RNA modifications are discussed.

## 3. Adenosine-Derived Modifications

The most common adenosine-derived modifications are m1A, m6A, 2${}^{\prime }$-O-methyladenosine (Am), N6,2${}^{\prime }$-O-dimethyladenosine (m6Am), and I (Figure 1).

The m1A modification occurs in different types of RNA, including tRNA, ribosomal RNA (rRNA), messenger RNAs (mRNAs), and long non-coding RNA (lncRNA) [48]. In tRNA, m1A is most commonly found at positions 9 and 58 (Figure 4), with the latter having been found in all domains of life [49]. Methylation at these positions protects and stabilizes the tRNA structure and facilitates a correct RNA folding [6,49]. m1A has also been observed at tRNA positions 14 (only in mammalian tRNA phenylalanine) and at position 22 (Figure 4) of bacterial tRNAs [49]. A is methylated to m1A in human tRNAs by the heterotetrameric TRMT6/TRMT61A (tRNA methyltransferase 6/61A) complex [48] within the cytosol and by the homodimeric TRMT61B methyltransferase in the mithocondria [50]. m1A modification has also been reported in mRNA and lncRNA, but here m1A seems to occur rarely and at very low stoichiometries [48,51,52]. In fact, mammalian mRNAs have an estimated m1A/A ratio ranging between 0.015% and 0.054% in cell lines and going up to 0.16% in mammalian tissues [51]. In particular, m1A modification appears to be highly enriched at 5’ UTR near the start-codon and occurs in GC-rich regions [19,51]. Importantly, the m1A modification is reversible and, in human cells, can be erased from tRNAs and mRNAs by the a-ketoglutarate-dependent dioxygenase alkB homolog 1 (ALKBH1) and ALKBH3, respectively [53]. Moreover, m1A profiles dynamically change in response to different cell stresses, such as glucose deprivation and heat shock [51,53]. Knockout mice for the ALKBH1 homolog exhibit severe embryonic phenotypes, supporting significant developmental functions [53]. Moreover, in HeLa cells, ALKBH1 knockout and knockdown experiments showed its involvement in regulating cell proliferation and translation by erasing the essential m1A58 of the initiator tRNA methionine, thus making this tRNA less available for translation initiation [53]. Methylation in m1A adds a positive charge to the modified adenosine that can alter RNA structure and affect protein-RNA interactions [51]. Moreover, m1A modification is placed on the Watson–Crick edge, where it disrupts the RNA base-pairing and induces local RNA duplex melting [54]. Thus, m1A modification interferes with the RT by causing misincorporation or truncation in cDNA synthesis, which makes this epitranscriptomic mark visible by HTS because sequencing reads show a drop of coverage or increased mismatch frequency when back-mapped to the reference genome [55].

The reversible m6A modification is the most prevalent post-transcriptional modification in mammalian mRNA and lncRNA [56]. Recently, Liu and co-workers, by using a refined (RIP)-seq (RNA immunoprecipitation followed by high-throughput sequencing) protocol, were able to identify between ∼10,000 and ∼20,000 peaks in polyadenylated RNAs and between ∼3000 and ∼7000 peaks in lncRNAs, across diverse human and mouse tissues [57]. The m6A/A ratio in humans is about 0.11–0.23%, and the m6A locations show tissue-specific regulation and evolutionary conservation [4,57]. m6A modification peaks are enriched for microRNA (miRNA) targeting sites [57], supporting their importance in gene regulation processes. Another characteristic of m6A is that SNPs are enriched within the m6A sites in all tissues, indicating a connection between m6A and risk for some diseases [57]. In mRNAs, m6A modifications are found at the DRACH motif (D=A/U/G, R=A/G, H=A/C/U), with GGACH being the most common variant [57,58]. A similar motif, DGACH (D=A/U/G, H=A/C/U), characterizes occurrence of some derivatives of the m6A modification, namely the N6-hydroxymethyladenosine (hm6A) and N6-formyladenosine (f6A) (Figure 5) [59]. While having been reported also within internal long exons [4], m6A occurs prevalently near stop-codons and 3’ UTRs (Figure 5) [5]. This well studied epitranscriptomic mark has been shown to play important roles in maintaining the circadian rhythm, regulating stem cell differentiation, and mediating response to many stress stimuli [56]. Even though m6A is located on the Watson–Crick edge, it does not cause mispairing; thus, its information gets erased by RT, and it is silent in HTS standard methods. For this reason, despite being known for decades and many hints of biological relevance, studies investigating m6A function flourished only recently, following the discovery of ad-hoc antibodies empowering immunoprecipitation-based high-throughput techniques for transcriptome-wide profiling [43,44].

Am modification and its derivative m6Am are both observed in polyadenylated RNAs (Figure 5), but Am is also found in rRNAs, small nuclear RNAs (snRNAs), small nucleolar RNAs (snoRNAs), and tRNAs at position 4 (Figure 4a). Am modifications are enriched in mRNAs at the AGAUC motif (Figure 5) [8,60], while m6Am modifications are enriched in snRNAs and mRNAs at the BCA motif (B=C/G/U) (Figure 5) [57]. The m6Am/A ratio in total RNAs from human tissues ranges from 0.0036% to 0.0169% [57]. Both Am and m6Am are reversible modifications that are erased by the Fat mass and obesity-associated protein (FTO) demethylase [61], which exhibits a preference for erasing methylation at m6Am [62]. Of note, even though m6Am can be derived from both Am or m6A, when it occurs at the first nucleotide after the m7G cap, it is always derived from Am [62], given that the first nucleotide of mRNAs after the cap is always either Am or A6Am, but never m6A (Figure 5). m6Am located at the first nucleotide position after the cap increases mRNA stability [62].

I is a substitutional RNA modification [63] found in all domains of life and produced by deamination of A (Figure 1) catalyzed by different enzymes in tRNA and other RNA species. Two main classes of deaminases catalyze this reaction in eukaryotes: the Adenosine Deaminase Acting on tRNA (ADAT) family [64] and the metazoan-restricted family of Adenosine Deaminases Acting on RNA (ADAR), the latter being responsible for A-to-I editing in transcripts that can form double-stranded RNA (dsRNA) beyond a certain length (∼15bp) [65], such as rRNAs, mRNAs, lncRNAs, and miRNA precursors [65,66,67]. Since A-to-I editing can change base pairing properties of the target nucleoside, this modification can have a profound effect on RNA structure and function. For the same reason, the I modification can be detected in standard RNA-seq experiments based on increased proportion of A-to-G mismatches at known sites of modification when aligned to the reference genome [24], even if a specific sequencing protocol based on selective reaction of inosine to a chemical pre-treatment has been developed to increase detection efficacy [25,68]. In tRNAs, I occurs at position 34 (Figure 4) in 8 out of 16 eukaryotic tRNAs with ANN anticodon (namely, the AGC, ACG, AAT, AAG, AGG, AGA, AGT, and AAC anticodons) where it is introduced by the ADAT2-ADAT3 heterodimer [65,69,70]. I34 also occurs in bacteria in tRNA Arginine (ACG anticodon) and tRNA Leucine (AAG anticodon), where it is introduced by the tRNA adenosine deaminase A (TadA) homodimer, while it is not present in archaea [70]. Position 34 is the first base of a tRNA anticodon (spanning positions 34–36 in the tRNA sequence), and it is named the “wobble” position due to involvement in flexible base pairing with the third position of a codon in the “codon-anticodon duplex” [7]. I34 is implicated in anticodon-codon recognition but it is still not clear whether its effect is to expand or reduce the number of codons a tRNA can recognize [64]. However, yeast knockouts for the ADAT2 or ADAT3 homologs (Tad2 and Tad3, respectively) are lethal, suggesting that I34 plays essential functions [64]. In mRNA, A-to-I editing occurs extensively in the transcriptome, especially in the brain, and it has been linked to stem cell differentiation, animal development and nervous system function [68]. In mRNAs’ untranslated regions, I shows increased frequency at 3’UTR compared to 5’UTR, maybe due to a higher number of dsRNA substrates for ADAR deamination in 3’UTRs [65]. A-to-I editing in 3’UTR has been linked to regulation of target gene expression due to creation of binding sites for miRNAs or RBPs [71,72]. A-to-I editing has been also observed in coding regions of mRNAs, despite this could introduce a nonsynonymous substitution and alter coding properties [65]. In fact, the translational machinery generally interprets I as its analogue G (Figure 1), even if rare and context-dependent decoding of I as A or U have also been reported [73]. Interestingly, exonic occurrence of A-to-I editing has been found to occur at multiple sites in the transcriptome of cells of the nervous system, particularly neurotransmitter receptor transcripts [65]. Supporting important roles in neuron function, dysfunction of ADAR genes has been related to several neurological disorders, including Amyotrophic lateral sclerosis (ALS) [65,74]. To date, millions of I sites have been detected, of which over 95% are found within Alu-elements in non-coding RNAs [35,75,76]. Vertebrates have three highly conserved ADAR genes: ADAR1, ADAR2, and ADAR3 [67]. Of them, ADAR1 and ADAR2 are ubiquitously expressed and competent for A-to-I deaminase activity, while the enigmatic ADAR3 is apparently a catalytically inactive family member characterized by brain-restricted expression in humans [67].

m1I, which derives from methylation of I, has only been found in archaea at tRNA position 57 (m1I57) and in the eukaryotic tRNA Alanine at position 37 (m1I37) [64,69]. Moreover, m1I57 can be further methylated into m1Im57 in thermophilic organisms [77]. Similarly to I, m1I can be detected by HTS because it alters base-pairing during cDNA synthesis [23,24]. Yeast knockouts for the enzyme responsible of the first step in m1I37 formation (i.e., Tad1, homolog of human ADAT1) are viable and do not show any gross phenotype, suggesting that this RNA modification does not play essential functions, at least under normal growth conditions [64].

## 4. Cytidine-Derived Modifications

Among the RNA bases, C has the lowest number of known modifications (Figure 2a and Figure 3b). Of them, the most common are m3C, m5C, hm5C, ac4C, and 2${}^{\prime }$-O-methylcytidine (Cm) (Figure 1), the first three of which have been observed in all domains of life [78].

C is methylated to m3C in tRNAs at position 47b and 32 (Figure 4b) by the methyltransferase-like 2 and 6 (METTL2, METTL6), while by METTL8 in mRNAs [79]. To date, biological functions associated with m3C modification have not been identified yet. Interestingly, it has been shown that this modification can be detectable by HTS without chemical pre-treatment [23].

m5C modification is an abundant modification in polyadenylated RNA, tRNA, rRNA, and viral RNA (Figure 4b and Figure 5) [80]. Despite m5C having been observed in all domains of life, there are notable differences concerning specific RNA types, such in the case of m5C occurrence in tRNA and mRNA that has been found in eukaria and archaea but not in bacteria [80]. In eukaryotes, m5C modification is documented in various tRNA positions, such as the 34, 38, 40, 48, 49, 50, and 72 (Figure 4b) [81]. Methylation of tRNA position 38 seems involved in regulating tRNA cleavage in response to thermal and oxidative stress [82,83]. In general, m5C modification appears involved in stabilizing the tRNAs, promoting protein synthesis [84] and regulating immune response [85]. In particular, Blanco and co-workers investigated the mechanisms underlying the observed interplay between the occurrence of tRNA methylation at cytosine-5 and translation [84,86,87]. Specifically, they found that hypomethylation of tRNAs caused by loss of the cognate RNA methyltransferase, which is the NOL1/NOP2/Sun domain family member 2 (NSUN2), facilitates endonucleolytic cleavage by angiogenin and consequent accumulation of 5${}^{\prime }$ tRNA fragments, which in turn inhibit cap-dependent protein synthesis [86]. Furthermore, they found that levels of m5C methylation in tRNA can modulate stem cell fate by regulating rate of protein synthesis since maintenance of stemness requires low levels of translation [87]. Formation of m5C is catalyzed by one of seven members of the NOP2/Sun domain family (NSUN1–NSUN7) of RNA methyl-transferases, as well as by the DNA methyltransferase 2 (DNMT2) enzyme [88]. Moreover, two m5C reader proteins are known: the Aly/ REF export factor (ALYREF) regulating the nuclear export of modified mRNAs [89] and the recently discovered Y-box binding protein 1 (YBX1) promoting bladder cancer by enhancing stability of target mRNAs [90]. In a recent work, Huang and co-workers developed an improved statistical method to investigate m5C methylome in human and mouse tissues and cell lines [91]. Importantly, they found evidence that tRNA methyltransferase NSUN2 plays a key role in writing the m5C mark in mRNA and that NSUN2-dependent m5C sites in mRNA show over-representation of a motif that resembles both sequence and structural features recognized by NSUN2 in tRNA [91].

5-hydroxymethylcytidine (hm5C) is a derivative of m5C that has been found in polyadenylated RNA and in rRNA. Its occurrence in all domains of life suggests that hm5C may play important regulatory roles [92]. In the protein-coding regions of mRNA, hm5C is found in the UCCUC repeats (Figure 5) and is enriched especially in the brain, where it promotes translation by restoring the translation efficiency of methylated substrates [21].

Cm modifications, similarly to all other Nm modifications (Am, Gm, Um), are found in many types of RNA, such as snRNA, snoRNA, and lncRNA [60]. In tRNA, they are often located at position 4 (Figure 4), which is usually an unmodified position with the exclusion of methylation of the 2${}^{\prime }$ hydroxyl of the ribose moiety [93]. Function of Nm modifications in tRNA remains unclear, but some suggestions of a role in translation have been indicated [93]. In mRNAs, Cm and other Nm modifications are enriched at the AGAUC motif, especially in the protein-coding regions, but also in 5’UTRs and 3’UTRs (Figure 5) [60].

N4-acetylcytidine (ac4C) is a modification found in all domains of life in mRNA, tRNA, and rRNA. In mRNA, acetylation of cytidine to ac4C is enriched in the coding region near the translation start-site, and in the 5’ UTR (Figure 5), where it is catalyzed by the acetyltransferase NAT10 [78]. In tRNA, it is located at position 12 and 34 (Figure 4b) [78,94]. Both in tRNA and mRNA, ac4C modification stabilizes the RNAs and promotes translation efficiency and gene expression [78].

Agmatidine (C+) modification in archaea and 2-lysidine (k2C) modification in bacteria are known to cause mispairing at least in codon recognition. Both C+ and k2C are found in the wobble position 34 of tRNA (Figure 4b), in archaea and bacteria, respectively, where they have a role in increasing and controlling the diversity of codon recognition [95,96]. At this wobble position, both C+ and k2C mostly pair with A instead of G, causing the tRNA to recognize the isoleucine AUA codon instead of the methionine AUG codon [95,97]. The similar function and position of C+ and k2C suggest that, in different species, different modifications at the same tRNA site may indeed be involved in a similar biological function.

## 5. Guanosine-Derived Modifications

In the landscape of epitranscriptomic marks, G-derived modifications vary the most in their complexity, ranging from very simple base methylations (involving only 1 stage of chemical modification) to very complex derivatives (requiring up to 9 stages) (Figure 1 and Figure 2b). Among them, most common G-derivatives are m1G, 2-methylguanosine (m2G), m2,2G, m7G, queuosine (Q), and o2yW. The complex biosynthetic pathway leading to formation of o2yW entails nine intermediate wyosine-family members that function as substrate for the next product. These ancient wyosine-family modifications are exclusively found in tRNA phenylalanine (anticodon GAA) of archaea and eukarya at position 37 (Figure 4c), where they play a role in reading-frame maintenance [98,99]. At this position, o2yW modification has been shown to cause RT misincorporation, but the underlying mechanism is not well characterized [23].

m1G at position 37 in tRNA (m1G37) (Figure 4c) is the precursor of 12 different modifications (Figure 2b), including the wyosine-family of modifications, and constitutes itself a widespred methylation found across all domains of life in a subset of tRNAs (namely, tRNA arginine, tRNA leucine, tRNA proline, and tRNA histidine) [98,100]. The enzyme tRNA m1G37 methyltransferase 5 (Trm5) responsible for m1G37 formation in eukaryotes and archaea is evolutionary unrelated to the bacterial counterpart TrmD [101]. m1G is located on the Watson–Crick edge and causes altered RT incorporation patterns, being often sequenced as T [23].

m2G and its derivative m2,2G are both found in tRNAs at positions 10 and 26 (Figure 4c), where they function in controlling and stabilizing tRNA structure, even though their biological roles are not well understood [102,103]. As well as in tRNAs, m2G modifications occur in snRNAs and rRNAs [8]. Both m2G and its derivative m2,2G involve RT incorporation [23]. The m2G causes RT misincorporation only slightly, while the mechanism of the m2,2G preventing base-pairing, especially in tRNA position 26, is particularly well characterized [23,102]. The m2,2G at position 26 stabilizes the tRNAs by forming a loop with the near nucleotides, including the m2G in position 10 [102]. The presence of two methyl groups in this loop prevents m2,2G from pairing with C; thus, it can pair only with A, U, or G [102] and is detectable in HTS as T with varying amounts of C and A [23].

m7G modification occurs both in mRNA and tRNA. In tRNA, it is found at positions 34, 36, and frequently at position 46 (Figure 4c), where it stabilizes tRNA structure [104]. In mRNA, m7G is enriched in coding regions, especially near start and stop codon, and at the 3’UTR in GA-rich motifs (Figure 5) [105].

Q modifications are irreversible post-transcriptional RNA modifications that occur at the GUN-anticodon sequence of tRNA (tRNAs for amino acids tyrosine, asparagine, aspartic acid, and histidine), where the modified G lies at the wobble position [106,107,108] and protects tRNAs from cleavage [109]. Q modification is only found in bacteria and eukarya, while, in archaea, a related modification, the archaeosine (G+), occurs in tRNA at position 15 [106,107]. The biosynthetic pathways leading to formation of Q in bacteria and of G+ in archaea both start with synthesis of the 7-Cyano-7-deazaguanosine (preQ0) intermediate from Guanosine 5’-triphosphate (GTP) catalyzed by the GTP cyclohydrolase I enzyme, while eukaryotes rely on import of the free base queuine from diet or intestinal flora [106,107,108,110]. In archaea, G+ is formed directly from preQ0, while, in bacteria, the formation of Q requires the following further steps: preQ0 is first converted to 7-aminomethyl-7-deazaguanosine (preQ1) in an NADPH-dependent reaction catalyzed by the preQ0 oxidoreductase; then, the enzyme tRNA-guanine transglycosylase inserts preQ1 into tRNA where the next steps of Q biosynthesis occur, such as the formation of epoxyqueuosine (oQ) catalyzed by the tRNA ribosyltransferase-isomerase and reduction of the epoxide catalyzed by the oQ reductase to finally form Q [106,111]. In mammals, Q can be further modified to form galactosyl-queuosine (galQ) and mannosyl-queuosine (manQ), while, in bacteria, it can be modified into glutamyl-queuosine (gluQ) [108,112]. In eukaryotes, Q modifications have been found to play a role in many essential functions, such as development, metabolism, and aging, and decreased levels of Q-modification of tRNA with GUN anticodon have been observed in many tumors and cancer cell lines [108].

## 6. Uridine-Derived Modifications

Uridine-derived modifications occur mostly in tRNA. Among RNA bases, uridine exhibits the largest variety of derivatives. Among them, 2-thiouridine (s2U) and 5-carbamoylmethyluridine (ncm5U) (Figure 1) in turn can be further modified to form most of the different uridine-derived RNA modifications (Figure 3c). Other abundant and widely studied uridine-derived modifications are Ψ and D (Figure 1), also commonly occurring in tRNA. Most U-derived modifications occur at the wobble position of tRNA, where they stabilize tRNA structure and promote protein synthesis [113,114]. s2U is formed by thiolation of U, that in eukaryotes is catalyzed by the tRNA thiouridase 1 and 2 complex (Ctu1-Ctu2) in the cytosol and by the mitochondrial thiolase (Mtu1) in mitochondrial tRNA [115], of which the latter retains highest structural similarity to the bacterial 2-thiouridylase MnmA (MnmA) enzyme compared to the former [115]. Occurrence of s2U at the wobble position has been involved in regulating the immune response [85].

Ψ is often referred to as the fifth RNA base, since it constitutes the most frequent post-transcriptional RNA modification [1]. This common epitranscriptomic mark has been identified in rRNA, snRNA, snoRNA, tRNA, and mRNA (Figure 5) [8,116]. Formation of Ψ is catalyzed by a family of related and evolutionarily conserved enzymes called pseudouridine synthases (PUSs), that encompasses two categories: one acting as “stand-alone“ enzymes and the other, restricted to eukarya and archaea, that acts via an RNA-dependent mechanism and utilizes one of a multitude of antisense box H/ACA small nucleolar RNAs (snoRNAs) as a guide RNA complementary to the sequence to be modified [117,118]. In tRNA, Ψ is conserved at position 55 in the T-arm loop (thus also known as the TΨC loop), where it contributes to tertiary structure by interacting with the loop of the D-arm [83]. Additionally, Ψ has been reported in about twenty other positions along the tRNA primary structure, especially in the anticodon stem (Figure 4d) [83,96,119,120,121,122]. Presence of Ψ modification has been linked to control of tRNA stability and structure [119,123]. In yeast and human mRNA, Ψ has been recently discovered as a widespread and conserved phenomenon, thanks to detection by HTS made possible by pre-treatment with N-cyclohexyl-N${}^{\prime }$-(2-morpholinoethyl)carbodiimide metho-p-toluenesulfonate (CMC), which acts as an RT block [124]. Of note, the predominant enzyme responsible for mRNA pseudouridylation in yeast, Pus1, acts as a “stand-alone” enzyme in which its target site of modification is determined by a structural motif [125].

1-methylpseudouridine (m1Ψ), a derivative of the Ψ, is located both on mRNAs and tRNAs. In mRNA, m1Ψ modification causes ribosome pausing and alters translation [126]. Moreover, it enhances human cell viability and appears involved in immune response [127].

D is exclusively conserved in tRNAs at positions 16, 17, 20, 20A, 20B, and 47 (Figure 4d), where it promotes tRNA flexibility [128,129]. Levels of D modification are found to be increased in cancer tissues [129]. D is detectable by HTS due to its ability to induce RT mismatches [23].

## 7. Epitranscriptomics Role in Physiology and Disease

The about 160 types of RNA modifications identified so far in tRNAs, rRNAs, mRNAs, snRNAs, and other RNAs have the potential to expand the chemical and structural properties of the four standard RNA bases, thus influencing the overall structure, interaction with proteins, and biological functions of the RNAs that carry them. Many of these co-transcriptional and post-transcriptional RNA modifications are reversible and dynamically regulated in response to various stimuli, enabling pools of variably modified molecules [51,130]. Relevant to human health, there is a growing list of diseases that have been associated with altered RNA modifications or to dysregulation of the enzymes responsible for writing, reading, or erasing them. Several excellent reviews have thoroughly described roles of epitranscriptomic marks in physiology and disease [96,131,132,133,134,135,136,137,138,139]. Here, we provide an overview of major molecular functions, biological processes, and human diseases associated with altered RNA modification patterns and/or misfunction of cognate enzymes that have emerged from the study of chemical modifications across diverse RNA classes.

In eukaryotes, the vast majority of known RNA modifications occur in tRNAs (i.e., 51 out of 66 listed in the RNA Modification Database at the time of writing) [140], and human tRNAs harbor, on average, 13 modified residues out of the 76 total residues that make up the canonical molecule [141]. Some of these modifications are common to most tRNAs (such as D at the D-loop or Ψ at the TΨC-loop), while others are tRNA-specific or limited to a subset of tRNAs (such as taurine-derived modifications restricted to a subset of mitochondrial tRNAs). Most tRNA modifications occur at the wobble position (position 34) and adjacent to that, at position 37 (Figure 4), where they enable the wobbling phenomenon and stabilize codon-anticodon interaction to prevent translational frameshifting [142]. The tRNAs are primarily known as critical adapters during protein synthesis to ensure proper translation of the RNA-copy of a protein-coding gene into the sequence of corresponding amino acids. However, this view has recently broadened to accommodate evidence of tRNA roles beyond translation, such in the case of tRNA fragments participating in cell stress response and regulating translational programs and the pace of protein synthesis [86,87,143,144]. Post-transcriptionally laid modifications have been shown to influence tRNA stability (such as the m1A, m7G, m5C, and D modifications), folding (m2,2G, Ψ, and D), translational efficiency (m1G), and global rate of protein synthesis (m5C) [145]. In general, tRNA modifications are essential for function and hypomodified tRNAs are targeted to degradation pathways, with specific modifications exerting a more dramatic effect than others maybe due to functional redundancy of some epitranscriptomic marks [146]. Dynamic control of tRNA modifications can provide a crucial regulatory mechanism participating in the cell response to environmental stimuli by modulating speed of ribosomal translation and promoting coordinated expression of subsets of functionally related genes [147,148,149]. In agreement with participating in core cell functions, defects in tRNA modifications and/or in the enzymes responsible for writing, reading or erasing these modifications have been associated with several human diseases, including different types of cancer, neurological, and mitochondrial-linked disorders [131]. Among others, proper tRNA modification at the wobble position appears to be crucial for health, and its loss has been linked to many different human diseases. For instance, the alkB homolog 8, tRNA methyltransferase (ALKBH8), that is required for the final step in the biogenesis of 5-methoxycarbonylmethyluridine (mcm5U) and for intermediate steps of other U-derived modifications at position 34 of several tRNAs, is up-regulated in bladder cancer [150] and inactivating mutations in its gene cause intellectual disability [151]. Notably, intellectual disability has also been linked to mutations in the genes of several further enzymes responsible for tRNA modification [152], including genes modifying the wobble position, such as ADAT3 [153], NSUN2 [154], and ELP2 (the elongator acetyltransferase complex subunit 2) [155]. Of them, mutations in NSUN2 have also been linked to the Dubowitz syndrome [156], the Noonan-like syndrome [157], and to different cancer types [158], while ELP2 is one subunit of the Elongator complex that is involved in U34 modifications of a subset of tRNAs and in which misfunction due to mutated components has also been linked to familial dysautonomia, amyotrophic lateral sclerosis, asthma, and cancer [152]. Moreover, the conserved tRNA thiolase acting on U34 composed of cytosolic thiouridylase subunit 1 and 2 (CTU1/2) have been shown to be up-regulated in human breast cancers and to sustain metastasis [159]. Finally, defects in taurine modifications (such as τm5U and its 2-thiouridine derivative τm5s2U) at the wobble position of mitochondrial tRNAsLeu(UUR) and mitochondrial tRNALys have been causally related to MELAS (mitochondrial encephalopathy, lactic acidosis, and stroke-like syndrome) and MERRF (myoclonus epilepsy with ragged-red fibers) syndromes, respectively [11].

Over a dozen different epitranscriptomic marks have been annotated in eukaryotic mRNAs [140] (Figure 5). Of note, many of these modifications seem not limited to protein coding mRNAs but shared among polyadenylated (poly-A) transcripts, including most recently appreciated lncRNAs [160]. These RNA modifications have been reported to play critical roles in several aspects of RNA metabolism, including maturation [59,161,162,163], export to the cytoplasm [164], efficiency and fidelity of translation [165,166,167,168,169], and RNA stability [170]. Accordingly, knockout experiments targeting key modification enzymes showed extreme phenotypes in mouse experiments, such as pre- or peri-natal lethality, supporting the notion that epitranscriptome plays fundamental and broad functions in mammals [171]. In general, internal modifications of mRNAs and lncRNAs may alter transcript secondary structure, thus promoting or inhibiting interaction with specific proteins [161]. Importantly, ability to influence RNA metabolism offers a strategy for fast and coordinated regulation of a set of transcripts in response to specific cell stresses, such in the case of m6A modification within 5’UTR promoting cap-independent translation in response to heat shock [166]. Methylation of adenosine to m6A is the prevalent chemical modification found in poly-A RNAs. In mRNAs, distribution of m6A is often biased towards stop codons and in the 3’UTR [5]. This epitranscriptomic mark has been shown to play a role in transcript maturation, nuclear export, enhancing translation efficiency and, depending on the specific reader protein, either increasing (when interacting with YTHDF1, the YTH-domain family member 1) or decreasing transcript stability (when interacting with YTHDF2) [145]. In overexpression or knockout mouse models, m6A mark and its related writer (METTL3, the methyltransferase like 3 and METTL14), reader (YTHDF1, YTHDF2, IGF2BP1, the insulin like growth factor 2 mRNA binding protein 1, IGF2BP2 and IGF2BP3), and eraser (FTO and ALKBH5, the alkB homolog 5) proteins [172] have been shown to take part in a plethora of cell and developmental processes, including adipogenesis [173], fertility [164], dopaminergic signaling in mid-brain [174], and stem cell self-renewal [175]. Coherently, altered m6A patterns have been linked to many human diseases, including different cancer types (reviewed in Reference [176]). Other frequently occurring chemical modifications in poly-A transcripts are the m1A, m5C, and Ψ modifications, which have been reported to be involved in nuclear export (m5C), enhancement of translation initiation (m1A), and modulation of translation (m5C), promoting RNA stability (Ψ) [80,145]. In addition, for these modifications, there is a growing list of annotated writer, reader, and eraser proteins paralleled by a growing list of human diseases associated with their mutation and misfunction, resulting in altered RNA modification patterns [145]. In eukaryotic rRNAs, more than 20 different RNA modifications have been documented so far [140]. In general, rRNA modifications seem to be mainly devoted at ensuring translation efficiency and accuracy by stabilizing proper folding of scaffold rRNAs in the ribosome [177]. However, it has been suggested that rRNA modifications may play roles beyond structural stability and provide a source of ribosomial heterogeneity useful to respond to environmental and developmental cues (reviewed in Reference [137]). Finally, RNA modifications and editing events are becoming more and more documented also in small RNAs, including miRNAs, Piwi-interacting RNA (piRNAs), and tRNA-derived small RNAs (tsRNAs), where they have been shown to participate in important processes, such as miRNA biogenesis, small RNA stability, and human host–pathogen responses, as well as to be associated with human disease states when dysfunctional (reviewed in Reference [178]).

## 8. Epitranscriptomics Web Resources

Many different databases currently provide systematic information about known RNA modifications and editing events or their transcriptome-wide annotated sites reported in different species (Table 1a). In particular, DARNED (DAtabase of RNa EDiting) is a database of RNA editing events (mostly A-to-I but also a few C-to-U editing events) and currently includes annotated sites for human, mouse, and fruit fly transcripts [179]. RADAR (a rigorously annotated database of A-to-I RNA) is another database collecting A-to-I modification sites, along with manually-curated annotations in humans, mice, and fruit flies [180]. Currently, the most comprehensive database of RNA editing events is the REDIportal [76], including over 4.5 million human A-to-I sites collected in 55 body sites of 150 healthy individuals from the Genotype-Tissue Expression (GTEx) project (https://gtexportal.org/), recently expanded to include editing events across different mouse brain tissues reported in a recent study [181]. The RNA Modification Base (RMBase) database (currently at version 2.0) stores modification sites concerning different epitranscriptomic marks (including m1A, m6A, Nm, Ψ, and m5C) collected from 47 studies among 13 species (such as human, rhesus monkey, chimpanzee, mouse, rat, pig, zebrafish, yeast and fission yeast, fruit fly, *Arabidopsis thaliana*, *Pseudomonas aeruginosa*, and *Escherichia coli*) [182]. Transcriptome-wide predicted m6A sites are also available from the MethylTranscriptome database (MeT-DB) that in the current version (version 2.0) includes data for 7 species (such as human, mouse, pig, yeast, fruit fly, zebrafish, and *Arabidopsis thaliana*) from 26 independent studies [183]. WHISTLE is a database providing human m6A modification sites predicted based on a machine learning approach [184]. REDIdb (version 3.0) is a database of RNA editing events in plants, including modification sites for many different plant species [185], while the FairBase database includes a comprehensive collection of A-to-I editing sites in 6 filamentous fungal species [186]. The tRNAdb provides a compendium of tRNA sequences, along with annotated modification sites from over 500 species across archaea, bacteria, eukarya, and viruses [121]. Most recently, Sajek and co-workers presented the T-psi-C resource [187], an up-to-date database of tRNA sequences from many species (including few viral molecules) enriched with 2D and 3D structure information, other than annotation of modified bases. The RNA modification Database (RNAMDB) provides a comprehensive collection of RNA modifications from archaea, bacteria, and eukaria that can be navigated by original nucleoside or RNA source [140]. The small nucleolar RNA orthological gene (snOPY) database [188] provides comprehensive information about snoRNAs of C/D or H/ACA class and the target RNAs, such as rRNAs and snRNAs, of which they guide chemical modification (2’O-methylation or pseudouridylation of target RNAs for C/D and H/ACA snoRNAs, respectively). Finally, MODOMICS is the most comprehensive database providing information about known RNA modifications across multiple species and types of RNA molecules, including related modifying enzymes, as well as modification pathways and position with respect to the RNA molecule carrying the given modification [8].

Many tools have been developed to computationally predict sites of modification for best characterized epitranscriptomic marks and are available as webtools or standalone software (Table 1b). Among them, the PACES webtool is a predictor of ac4C sites in human mRNA sequences [189]. The GIREMI [190], and RNAEditor [191] webtools are available to predict A-to-I editing sites, while the REDITools [192] comprise a collection of scripts that can be downloaded and used to predict human A-to-I editing sites from next-generation sequencing data. The SRAMP [193] and RNAMethPre [194] webtools can be used to predict occurrence of m6A sites in human and mouse mRNAs and lncRNAs (SRAMP) or mRNAs (RNAMethPre), while the RNAm5CFinder webtool predicts m5C sites in human and mouse RNA sequences [195]. The PPUS webtool predicts Ψ sites specifically catalyzed by pseudouridine synthase (PUS) enzymes in human and yeast mRNA, tRNA, snoRNA, and rRNA sequences [196] Similarly, the iRNA-PseU [197] webtool predicts pseudouridylation sites in mRNA, tRNA, snoRNA, and rRNA sequences from humans, yeast, and mice. From the same group, the iRNA-m2G webtool predicts m2G sites in eukaryotic RNA transcript sequences based on nucleotide frequencies and conservation surrounding m2G sites in the training data set of tRNA sequences from humans, mice, and yeast [103]. The SnoSCAN algorithm [198] is provided as a webserver (http://lowelab.ucsc.edu/snoscan/) that predicts occurrence of Nm modification sites in rRNAs, as well as occurrence of guide snoRNA genes, that are responsible for Nm modification in rRNAs, in yeast, humans, and archaea [199]. The Lowe’s lab has similarly made available as a webserver tool (http://lowelab.ucsc.edu/snoGPS/) the snoGPS algorithm [199] to predict snoRNA genes, that guides pseudouridylation in rRNAs in yeast, humans, and archaea [200]. Finally, HAMR software predicts some RNA modifications (such as m1A and m3C) from RNA-seq data based on the statistical analysis of patterns of mismatches based on the hypothesis that certain modified ribonucleotides alter base pairing, leading to errors in cDNA synthesis during RNA-seq library preparation [23,201].

## 9. Conclusions

More and more, we appreciate contribution of post-transcriptional regulation to the complexity of gene-expression regulation. Different regulatory layers work complexly and dynamically across different types of cells to ensure a gene expression outcome finely tuned and consonant with cell state. Best characterized examples, such as m6A (6-methyladenosine), uncover key contributions of epitranscriptome in this complex interplay regulating cellular, developmental, and disease processes. Although we already know that RNA modification and editing events are involved in maintaining the functionality of many living systems by impacting on many essential RNA-processing events, such as splicing, transport, localization, translation, and stability, research on functions and mechanisms acted by the epitranscriptome is still in an early phase. Moreover, we cannot rule out different biological roles even for the same RNA modification occurring at the same site in different species. Accordingly, the list of human diseases linked to altered RNA modifications or caused by misfunction of related RBPs is growing and so is the awareness of medicine and pharmaceutics towards epitranscriptomics [171]. Recent advances in detection made available for some RNA modifications have renewed interest in the field and accelerated the pace of discoveries. However, further boost of the field is hampered by lack of transcriptome-wide assays for mapping the vast majority of the many yet enigmatic RNA modifications, as well as the ones likely to be discovered in the near future. Moreover, solid and standardized statistical and computational pipelines, together with orthogonal experimental assays, are needed to ensure accuracy, reliability, and reproducbility of high-throughput data sets produced and conclusions drawn from their analysis. To this end, it would be timely to set a large and coordinated international consortium project similar in scope and breadth of the ENCyclopedia Of DNA Elements (ENCODE) project [202]. In fact, similar to the essential empowering of epigenomics studies made by the ENCODE project, a similar coordinated effort could greatly accelerate development of new technologies suitable to profile diverse RNA modification types and their dynamics and setting of analytical best-practices to tackle false-positives and accuracy issues. Ideally new sequencing protocols will provide quantitative information about the extent of modifications, in many different cell types and possibly at single cell resolution. Here, we provide an organized census of currently known RNA modifications and related web resources. We believe that our proposal of a scheme to categorize this knowledge could be helpful to the field by providing a flexible framework that can simplify navigation of current information and integrate more RNA modifications as they emerge. Moreover, illustration of this scheme as base-specific or global tree-representations can be used to adjust breadth of view (from focusing on a single base and its modifications to the full set and their cross-talks) and allows to overlay variable levels of details, such as organisms, synthetic pathways, and interacting partners.

## Figures and Tables

**Figure 1 ijms-21-04684-f001:**
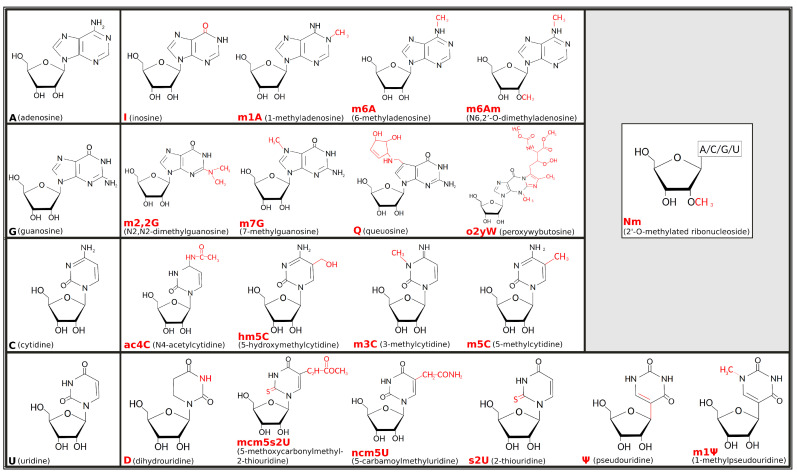
Chemical formulas for the main types of epitranscriptomic marks. Unmodified ribonucleic acid (RNA) bases are shown in black to the left-most position of each row, while chemical formulas to the right are some of their cognate modified forms, with chemical changes highlighted in red. Grey inset at top-right corner shows the 2’-O-methylation (or Nm, where N stands for any nucleoside), a common modification that can appear on any of the ribonucleosides.

**Figure 2 ijms-21-04684-f002:**
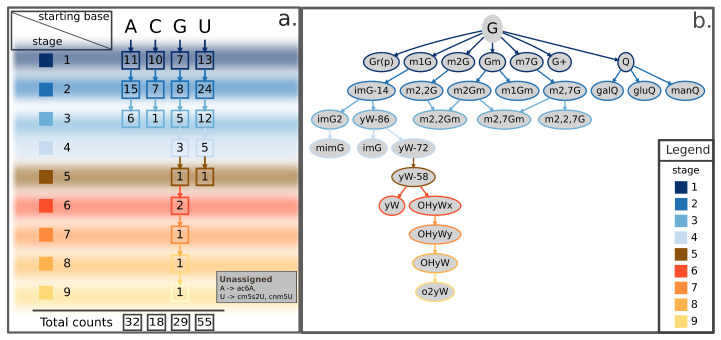
Summary scheme and representative tree of guanosine-derived RNA modifications illustrating the proposed categorization method. (**a**) For each ribonucleoside (A, C, G, U), the scheme reports in squares the number of known chemically-modified derivatives occurring at the given stage of modification, as well as its total counts (bottom). Stages of modification range from 1 (top-left) to 9 (bottom-left), each marked by a different color illustrated in the filled squares on the left and bordering squares that report cognate counts for each ribonucleoside. The scheme summarizes a proposed classification for 134 currently known chemical modifications according to their root nucleoside. Some of these modified RNA bases can be derived by means of one single chemical modification of each natural ribonucleoside (stage 1), some others can be obtained from one further step of chemical modification acting upon stage 1-products (stage 2), and so on up to the maximum number of modifying steps (stage 9). The bottom-right inset lists three additional RNA modifications—the A-derived ac6A and the two U-derived cm5s2U and cnm5U modifications—currently lacking enough details on the synthesis process to be assigned a stage in the scheme. (**b**) Tree-representation of all known G-derived RNA modifications following the classification method summarized in (a). Border colors of leaves (or nodes) in the tree representation indicate the corresponding stage according to the color-scheme reported in the legend. Shown RNA modifications are the union of current knowledge gathered from eukarya, bacteria, and archaea. A = Adenosine; ac6A = N6-acetyladenosine; C = Cytidine; cm5s2U = 5-carboxymethyl-2-thiouridine; cnm5U = 5-cyanomethyluridine; G+ = archaeosine; G = Guanosine; galQ = galactosyl-queuosine; gluQ = glutamyl-queuosine; Gm = 2${}^{\prime }$-O-methylguanosine; Gr(p) = 2${}^{\prime }$-O-ribosylguanosine (phosphate); imG = wyosine; imG-14 = 4-demethylwyosine; imG2 = isowyosine; m1G = 1-methylguanosine; m1Gm = 1,2${}^{\prime }$-O-dimethylguanosine; m2Gm = N2,2${}^{\prime }$-O-dimethylguanosine; m2,2Gm = N2,N2,2${}^{\prime }$-O-trimethylguanosine; m2,7Gm = N2,7,2${}^{\prime }$-O-trimethylguanosine; m2G = 2-methylguanosine; m2,2G = N2,N2-dimethylguanosine; m2,2Gm = N2,N2,2${}^{\prime }$-O-trimethylguanosine; m2,2,7G = N2,N2,7-trimethylguanosine; m2,7G = N2,7-dimethylguanosine; m2,7Gm = N2,7,2${}^{\prime }$-O-trimethylguanosine; m7G = 7-methylguanosine; manQ = mannosyl-queuosine; mimG = methylwyosine; o2yW = peroxywybutosine; OHyW = hydroxywybutosine; OHyWx = undermodified hydroxywybutosine; OHyWy = methylated undermodified hydroxywybutosine; Q = queuosine; U = Uridine; yW = wybutosine; yW-58 = 7-aminocarboxypropylwyosine methyl ester; yW-72 = 7-aminocarboxypropylwyosine; yW-86 = 7-aminocarboxypropyl-demethylwyosine.

**Figure 3 ijms-21-04684-f003:**
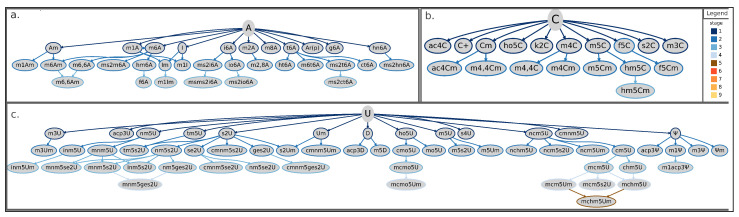
Tree-representation of known RNA modifications following the proposed categorization scheme. Starting from the initial base (adenosine in panel (**a**), cytidine in (**b**), and uridine in (**c**)), each connection (or branch) corresponds to a chemical modification added on the previous state (that is, the preceding leave). Border colors of leaves (or nodes) in the tree representation indicate the corresponding stage according to the color-scheme reported in the legend. Shown RNA modifications are the union of current knowledge gathered from eukarya, bacteria, and archaea. A = adenosine; ac4C = N4-acetylcytidine; ac4Cm = N4-acetyl-2${}^{\prime }$-O-methylcytidine; acp3Ψ = 3-(3-amino-3-carboxypropyl)pseudouridine; acp3D = 3-(3-amino-3-carboxypropyl)-5,6-dihydrouridine; acp3U = 3-(3-amino-3-carboxypropyl)uridine; Am = 2${}^{\prime }$-O-methyladenosine; Ar(p) = 2${}^{\prime }$-O-ribosyladenosine (phosphate); C+ = agmatidine; C = cytidine; chm5U = 5-carboxyhydroxymethyluridine; Cm = 2${}^{\prime }$-O-methylcytidine; cm5U = 5-carboxymethyluridine; cmnm5ges2U = 5-carboxymethylaminomethyl-2-geranylthiouridine; cmnm5s2U = 5-carboxymethylaminomethyl-2-thiouridine; cmnm5se2U = 5-carboxymethylaminomethyl-2-selenouridine; cmnm5U = 5-carboxymethylaminomethyluridine; cmnm5Um = 5-carboxymethylaminomethyl-2${}^{\prime }$-O-methyluridine; cmo5U = uridine 5-oxyacetic acid; ct6A = cyclic N6-threonylcarbamoyladenosine; D = dihydrouridine; f5C = 5-formylcytidine; f5Cm = 5-formyl-2${}^{\prime }$-O-methylcytidine; f6A = N6-formyladenosine; g6A = N6-glycinylcarbamoyladenosine; ges2U = 2-geranylthiouridine; hm5C = 5-hydroxymethylcytidine; hm5Cm = 2${}^{\prime }$-O-Methyl-5-hydroxymethylcytidine; hm6A = N6-hydroxymethyladenosine; hn6A = N6-hydroxynorvalylcarbamoyladenosine; ho5C = 5-hydroxycytidine; ho5U = 5-hydroxyuridine; ht6A = hydroxy-N6-threonylcarbamoyladenosine; I = inosine; i6A = N6-isopentenyladenosine; Im = 2${}^{\prime }$-O-methylinosine; inm5s2U = 5-(isopentenylaminomethyl)-2-thiouridine; inm5U = 5-(isopentenylaminomethyl)uridine; inm5Um = 5-(isopentenylaminomethyl)-2${}^{\prime }$-O-methyluridine; io6A = N6-(cis-hydroxyisopentenyl)adenosine; k2C = 2-lysidine; m1A = 1-methyladenosine; m1acp3Ψ = 1-methyl-3-(3-amino-3-carboxypropyl)pseudouridine; m1Am = 1,2${}^{\prime }$-O-dimethyladenosine; m1I = 1-methylinosine; m1Im = 1,2${}^{\prime }$-O-dimethylinosine; m1Ψ = 1-methylpseudouridine; m2A = 2-methyladenosine; m2,8A = 2,8-dimethyladenosine; m3C = 3-methylcytidine; m3U = 3-methyluridine; m3Um = 3,2${}^{\prime }$-O-dimethyluridine; m3Ψ = 3-methylpseudouridine; m4C = N4-methylcytidine; m4Cm = N4,2${}^{\prime }$-O-dimethylcytidine; m4,4C = N4,N4-dimethylcytidine; m4,4Cm = N4,N4,2${}^{\prime }$-O-trimethylcytidine; m5C = 5-methylcytidine; m5Cm = 5,2${}^{\prime }$-O-dimethylcytidine; m5D = 5-methyldihydrouridine; m5s2U = 5-methyl-2-thiouridine; m5U = 5-methyluridine; m5Um = 5,2${}^{\prime }$-O-dimethyluridine; m6A = 6-methyladenosine; m6Am = N6,2${}^{\prime }$-O-dimethyladenosine; m6,6A = N6,N6-dimethyladenosine; m6,6Am = N6,N6,2${}^{\prime }$-O-trimethyladenosine; m6t6A = N6-methyl-N6-threonylcarbamoyladenosine; m8A = 8-methyladenosine; mchm5U = 5-(carboxyhydroxymethyl)uridine methyl ester; mchm5Um = 5-(carboxyhydroxymethyl)-2${}^{\prime }$-O-methyluridine methyl ester; mcm5s2U = 5-methoxycarbonylmethyl-2-thiouridine; mcm5U = 5-methoxycarbonylmethyluridine; mcm5Um = 5-methoxycarbonylmethyl-2${}^{\prime }$-O-methyluridine; mcmo5U = uridine 5-oxyacetic acid methyl ester; mcmo5Um = 2${}^{\prime }$-O-methyluridine 5-oxyacetic acid methyl ester; mnm5ges2U = 5-methylaminomethyl-2-geranylthiouridine; mnm5s2U = 5-methylaminomethyl-2-thiouridine; mnm5se2U = 5-methylaminomethyl-2-selenouridine; mnm5U = 5-methylaminomethyluridine; mo5U = 5-methoxyuridine; ms2ct6A = 2-methylthio cyclic N6-threonylcarbamoyladenosine; ms2hn6A = 2-methylthio-N6-hydroxynorvalylcarbamoyladenosine; ms2i6A = 2-methylthio-N6-isopentenyladenosine; ms2io6A = 2-methylthio-N6-(cis-hydroxyisopentenyl) adenosine; ms2m6A = 2-methylthio-6-methyladenosine; ms2t6A = 2-methylthio-N6-threonylcarbamoyladenosine; msms2i6A = 2-methylthiomethylenethio-N6-isopentenyl-adenosine; ncm5U = 5-carbamoylmethyluridine; nchm5U = 5-carbamoylhydroxymethyluridine; ncm5s2U = 5-carbamoylmethyl-2-thiouridine; ncm5Um = 5-carbamoylmethyl-2${}^{\prime }$-O-methyluridine; nm5ges2U = 5-aminomethyl-2-geranylthiouridine; nm5s2U = 5-aminomethyl-2-thiouridine; nm5se2U = 5-aminomethyl-2-selenouridine; nm5U = 5-aminomethyluridine; s2C = 2-thiocytidine; s2U = 2-thiouridine; s2Um = 2-thio-2${}^{\prime }$-O-methyluridine; s4U = 4-thiouridine; se2U = 2-selenouridine; t6A = N6-threonylcarbamoyladenosine; tm5U = 5-taurinomethyluridine; tm5s2U = 5-taurinomethyl-2-thiouridine; U = uridine; Um = 2${}^{\prime }$-O-methyluridine; Ψ = pseudouridine; Ψm = 2${}^{\prime }$-O-methylpseudouridine.

**Figure 4 ijms-21-04684-f004:**
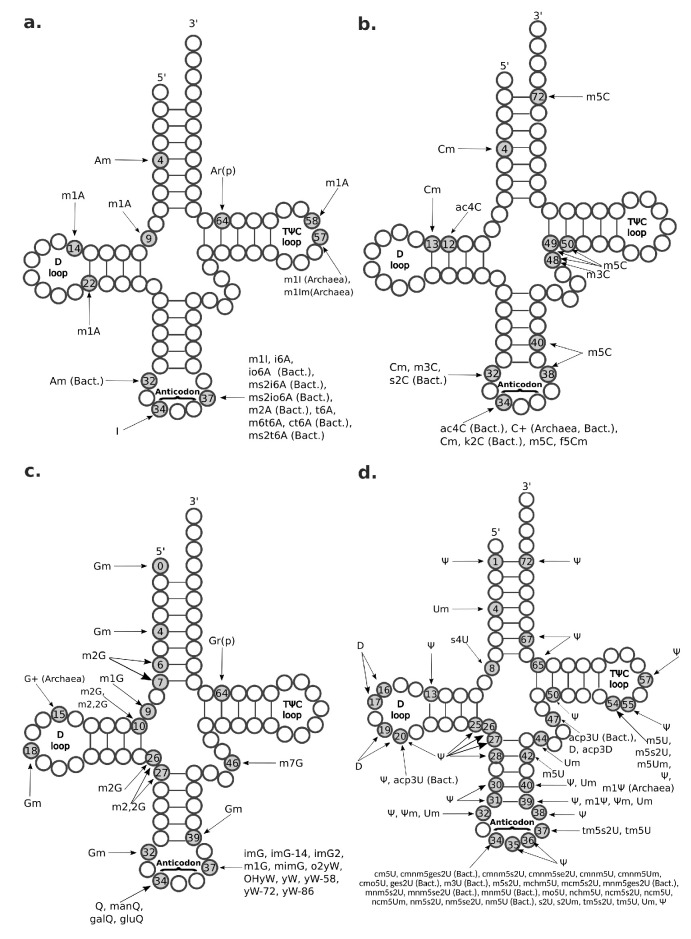
Known transfer RNA (tRNA) modifications. A schematic representation of the tRNA secondary structure is shown with circles representing RNA residues. Grey circles and numbers therein represent modified RNA residues and their position along the tRNA primary sequence. Connecting lines between RNA residues indicate base pairing. Three preeminent tRNA regions are labeled: the D-loop (residues 14–21), the anticodon (residues 34–36), and the TΨC-loop (residues 54–60). (**a**) tRNA modifications having as original substrate adenosine (A) residues; (**b**) tRNA modifications having as original substrate cytidine (C) residues; (**c**) tRNA modifications having as original substrate guanosine (G) residues; (**d**) tRNA modifications having as original substrate uridine (U) residues. ac4C = N4-acetylcytidine; acp3D = 3-(3-amino-3-carboxypropyl)-5,6-dihydrouridine; acp3U = 3-(3-amino-3-carboxypropyl)uridine; Am = 2${}^{\prime }$-O-methyladenosine; Ar(p) = 2${}^{\prime }$-O-ribosyladenosine (phosphate); bact. = bacterial; C+ = agmatidine; Cm = 2${}^{\prime }$-O-methylcytidine; cm5s2U = 5-carbamoylmethyl-2-thiouridine; cmnm5ges2U = 5-carboxymethylaminomethyl-2-geranylthiouridine; cmnm5s2U = 5-carboxymethylamino methyl-2-thiouridine; cmnm5se2U = 5-carboxymethylaminomethyl-2-selenouridine; cmnm5U = 5-carboxymethylaminomethyluridine; cmnm5Um = 5-carboxymethylaminomethyl-2${}^{\prime }$-O-methyluridine; cmo5U = uridine 5-oxyacetic acid; ct6A = cyclic N6-threonylcarbamoyladenosine; D = dihydrouridine; f5Cm = 5-formyl-2${}^{\prime }$-O-methylcytidine; galQ = galactosyl-queuosine; ges2U = 2-geranylthiouridine; gluQ = glutamyl-queuosine; Gm = 2${}^{\prime }$-O-methylguanosine; Gr(p) = 2${}^{\prime }$-O-ribosylguanosine (phosphate); I = inosine; i6A = N6-isopentenyladenosine; imG = wyosine; imG-14 = 4-demethylwyosine; imG2 = isowyosine; io6A = N6-(cis-hydroxyisopentenyl)adenosine; k2C = 2-lysidine; m1A = 1-methyladenosine; m1G = 1-methylguanosine; m1I = 1-methylinosine; m1Im = 1,2${}^{\prime }$-O-dimethylinosine; m1Ψ = 1-methylpseudouridine; m2,2G = N2,N2-dimethylguanosine; m2A = 2-methyladenosine; m2G = 2-methylguanosine; m3C = 3-methylcytidine; m3U = 3-methyluridine; m5C = 5-methylcytidine; m5s2U = 5-methyl-2-thiouridine; m5U = 5-methyluridine; m5Um = 5,2${}^{\prime }$-O-dimethyluridine; m6t6A = N6-methyl-N6-threonylcarbamoyladenosine; m7G = 7-methylguanosine; manQ = mannosyl-queuosine; mchm5U = 5-(carboxyhydroxymethyl)uridine methyl ester; mcm5s2U = 5-methoxycarbonylmethyl-2-thiouridine; mcm5U = 5-methoxycarbonylmethyluridine; mimG = methylwyosine; mnm5ges2U = 5-methylaminomethyl-2-geranylthiouridine; mnm5s2U = 5-methylaminomethyl-2-thiouridine; mnm5se2U = 5-methylaminomethyl-2-selenouridine; mnm5U = 5-methylaminomethyluridine; mo5U = 5-methoxyuridine; ms2i6A = 2-methylthio-N6-isopentenyladenosine; ms2io6A = 2-methylthio-N6-(cis-hydroxyisopentenyl) adenosine; ms2t6A = 2-methylthio-N6-threonylcarbamoyladenosine; nchm5U = 5-carbamoylhydroxymethyluridine; ncm5U = 5-carbamoylmethyluridine; ncm5Um = 5-carbamoylmethyl-2${}^{\prime }$-O-methyluridine; nm5s2U = 5-aminomethyl-2-thiouridine; nm5se2U = 5-aminomethyl-2-selenouridine; nm5U = 5-aminomethyluridine; o2yW = peroxywybutosine; OHyW = hydroxywybutosine; Q = queuosine; s2C = 2-thiocytidine; s2U = 2-thiouridine; s2Um = 2-thio-2${}^{\prime }$-O-methyluridine; t6A = N6-threonylcarbamoyladenosine; tm5s2U = 5-taurinomethyl-2-thiouridine; tm5U = 5-taurinomethyluridine; Um = 2${}^{\prime }$-O-methyluridine; yW = wybutosine; yW-58 = 7-aminocarboxypropylwyosine methyl ester; yW-72 = 7-aminocarboxypropylwyosine; yW-86 = 7-aminocarboxypropyl-demethylwyosine; Ψ = pseudouridine; Ψm = 2${}^{\prime }$-O-methylpseudouridine.

**Figure 5 ijms-21-04684-f005:**
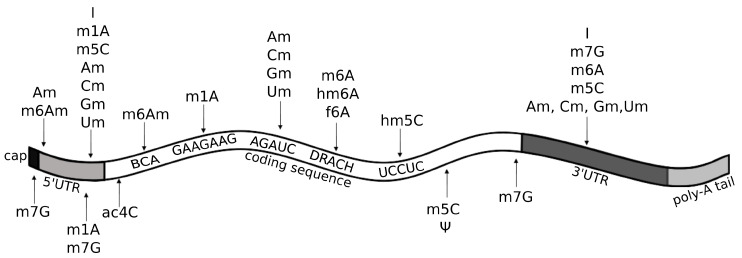
Known RNA modifications in messenger RNA (mRNA). The figure lists epitranscriptomic marks found in mRNA, along with their preferred location and motif at occurrence sites, if known. Of note, recent reports on Ψ and m5C marks in mRNA highlight the importance of structural motifs as determinants for modification (see text). A = adenosine; ac4C = N4-acetylcytidine; Am = 2’-O-methyladenosine; BCA motif = (B = C/G/U); C = cytidine; Cm = 2’-O-methylcytidine; DRACH motif (D=A/U/G, R=A/G, H=A/C/U); f6A = N6-formyladenosine; G = guanosine; Gm = 2’-O-methylguanosine; hm5C = 5-hydroxymethylcytidine; hm6A = N6-hydroxymethyladenosine; I = inosine; m1A = 1-methyladenosine; m5C = 5-methylcytidine; m6A = 6-methyladenosine; m6Am = N6,2${}^{\prime }$-O-dimethyladenosine; m7G = 7-methylguanosine; U = uridine; Um = 2’-O-methyluridine; Ψ = pseudouridine.

**Table 1 ijms-21-04684-t001:** Web resources for epitranscriptomics. (**a**) List of databases storing information on known RNA modifications and their sites of occurrence. (**b**) List of web resources providing prediction tools for best characterized RNA modifications.

Database	Description	Reference
DARNED	Database of A-to-I editing sites in humans, mice, and fruit flies	[179]
FairBase	RNA editing sites in fungi	[186]
MeT-DB	m6A sites in 7 species	[183]
MODOMICS	RNA modifications and pathways from multiple species	[8]
RADAR	Curated A-to-I editing sites in humans, mice, and flies	[180]
REDIdb	RNA editing sites in plants	[185]
REDIportal	A-to-I sites across different human tissues and mouse brain	[76]
RNAMDB	RNA modifications from multiple species	[140]
RMBase	Multiple RNA modification sites in 13 species	[182]
snOPY	snoRNAs, snoRNA gene loci, and target RNAs from multiple species	[188]
tRNAdb	RNA editing sites in tRNA in archaea, bacteria, eukarya, and viruses	[121]
T-psi-C	Up-to-date collection of tRNA sequences and structures from multiple species	[187]
WHISTLE	Predicted m6A sites in humans	[184]
**Tool**	**Description**	**Reference**
GIREMI	A-to-I sites in humans	[190]
HAMR	RNA modification sites based on RT-mutation profile	[23]
iRNA-m2G	m2G sites in humans, yeast, and mice	[103]
iRNA-PseU	Ψ sites in humans, yeast, and mice	[197]
PACES	ac4C sites in humans	[189]
PPUS	Ψ sites in humans and yeast	[196]
REDItools	A-to-I sites in humans	[192]
RNAEditor	A-to-I sites in humans and mice	[191]
RNAm5Cfinder	m5C sites in humans and mice	[195]
RNAMethPre	m6A sites in humans and mice	[194]
snoGPS	Pseudouridylation guide snoRNAs in humans and yeast	[199]
SnoSCAN	Nm sites in humans, yeast, and archaea	[199]
SRAMP	m6A sites in humans and mice	[193]

**Abbreviations:** A-to-I:adenosine to inosine; ac4C:N4-acetylcytidine; m5C = 5-methylcytidine; m6A = 6-methyladenosine; m2G = 2-methylguanosine; Nm = 2${}^{\prime }$-O-methylation; Ψ = pseudouridine; RT = reverse transcriptase.

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
