# Peer review of "A Census and Categorization Method of Epitranscriptomic Marks"

_ijms, 2020, doi:10.3390/ijms21134684_

Round 1
Reviewer 1 Report
This review focuses on the complex field of RNA modifications (epitranscriptomics). The first version of the review already read well and provided an interesting overview of the field, proposing a new classification scheme and framework for the study of epitranscriptomic marks. In this updated version, the authors have considerably improved the manuscript, notably in providing extra references and replying to every point suggested in the reviews. In particular, the discussion of third-generation sequencing technologies for detecting RNA modifications is a great addition to the text and makes the review more current. The new figures are also clearer, which is especially noticeable for Figure 1. Overall, this review is now clearer and quite comprehensive, and should be of great interest to a wide variety of readers from the RNA, epigenetic and chemical biology fields especially.
A minor detail: I am not sure if this is a problem on my end or an error which occurred during file format conversion, but I was unable to open Supplemental Figure 1.
Author Response
We thank the reviewer for the thorough revision. His/her valuable comments and suggestions helped us very much to improve our manuscript.
Point 1: A minor detail: I am not sure if this is a problem on my end or an error which occurred during file format conversion, but I was unable to open Supplemental Figure 1.
Response 1: We thank the reviewer for pointing out this issue. We regenerated the figure to fix possible file corruption.
Reviewer 2 Report
First of all, I would like to thank the authors for addressing all the concerns raised in the first round of revision. The manuscript is very comprehensive, focus on a very up-to-date topic and it is well written. It has all the requirements to be accepted for publication.
Author Response
We thank the reviewer for the positive comment. We are really pleased to have been able to treasure his/her valuable comments and thus improve our manuscript.
Reviewer 3 Report
The authors have fully addressed the previous comments of this Reviewer and I am fully satisfied with the outcome. In addition, the authors have added new information that I find valuable. I only have just three minor comments:
1- Line 281: there is a question mark after Reference 69.
2- In Line 489 the authors use the term "laying, reading or erasing"; people in the filed usually refer to these processes as "writing, reading or erasing". I suggest replacing "laying" for "writing"
3- The authors might want to add a reference to the database T-psi-C (Sajek et al Nucleic Acids Res 2020)
Author Response
We thank the reviewer for the appreciation of the work done and for his/her valuable comments that helped us reaching this positive result.
Point 1: Line 281: there is a question mark after Reference 69.
Response 1: We thank the reviewer for pinpointing this misprint. We regenerated the PDF and fixed the problem, that was due to incomplete processing of the bibliography by the reference management software.
Point 2: In Line 489 the authors use the term "laying, reading or erasing"; people in the filed usually refer to these processes as "writing, reading or erasing". I suggest replacing "laying" for "writing"
Response 2: We thank the reviewer for his/her suggestion. We changed the line 489 as recommended.
Point 3: The authors might want to add a reference to the database T-psi-C (Sajek et al Nucleic Acids Res 2020)
Response 3: Thank you for this suggestion: we added the T-psi-C resource and its reference paper in our manuscript. In particular, we added it to the list of resources presented in Table 1a and commented it at lines 572-575 in section 8 (Epitranscriptomics web resources) of our revised manuscript.
This manuscript is a resubmission of an earlier submission. The following is a list of the peer review reports and author responses from that submission.
Round 1
Reviewer 1 Report
The Review with the title "A census and categorization method of epitranscriptomic marks" describes in general the importance of epitranscriptomics to identify RNA modifications. THis is important for several species of cellular RNAs and the interconnection of writer, reader and eraser and their functions become more and more important.
Major points:
1.) I would suggest to state numbers like >1,500 RNA binding proteins as current catalogue also in the relation of the species. So is this the number in human, all mammals or in general also plants etc. So referring to OMIM seems to be human specific.
2.) In the introduction they start with the example of m6A but I would like to first get a paragraph about the different known modifications and which of them are currently detectable at all.
3.) The problematic is stated focusing on HTS but not mentioning beside the 2. NGS technologies 3. NGS technologies directly using RNA without transformation to cDNA. I wouldlike to see a overview of possible Modifications which can be seen via NGS.
4.) I wonder why the Review article is describing/ proposing a new method to classify create a nomenclature for RNA modifications. For this new approach I would think data to prove the efficiency are needed.
5.) On page 5 they stated mammalian mRNAs with >20.000 peaks annotated in human polyadenylated RNAs. Is it now mammalian or human specific?
6.) For the websites and tools tools like snoscan or snoop for methylation or pseudouridylation are missing? I found it hard to give a full overview also databases like snoopy contain inforamtion of modifications but then specifically for a kind of RNA species. SO this should be clearly stated and also if this are human specific databases or webtools in principle.
7.) In the end I miss the epitranscriptomics part and explanation of the creation of such data. In general the categorization is existing but could be more clearly categorized and expalined than at current state of the review article.
8.) The different Nucelotide derived modifications should be strucutred whether for species or modifcation group or something to make it like a compendium to look to find the modification of interest and then their explanation or examples or specific databases.
Reviewer 2 Report
In this review, the authors provide an overview of the complex and rapidly-expanding field of RNA modifications (epitranscriptomics), a very interesting and broad-ranging topic. Furthermore, the authors propose a categorization method for these marks based on the enzymatic steps needed for their biosynthesis, a reasonable approach. Overall, this is an interesting review about an important subject, which reads well and cites a broad set of the literature.
While the categorization approach is clear from the text it was a bit less from Fig 1, which could probably be improved. It could help to show at least a partial categorization ‘tree’ in Fig 1, maybe in-between panels (a) and (b), which, although not comprehensive, would make clear the idea behind Fig 1A and could also highlight some of the better-studied marks, such as m6A and ψ (this could be just a partial tree from Fig 2). In Fig 1A it is also unclear (and not explained in the legend) why phosphates and queuosines are placed at the bottom (does it put them in the “unassigned” category?), nor their number (is that the number of steps necessary for their biosynthesis?). This should be explained in the legend rather than just in the text later (lines 149-150). It would also be very useful to include as a supplemental figure a ‘complete’ tree for Fig 1A (with modification names, in the style of Fig 2).
One feature (which could be seen either as an advantage or as a disadvantage) of this categorization scheme is that it visually stresses marks that require a large number of biosynthetic steps, rather than their prevalence (in Fig 1A); at our present state of understanding most of these marks are tRNA modified bases. It could be useful to use a color-code in Fig 1A to estimate how many of these classified bases are tRNA-exclusive vs. the others (or alternatively color-code the potential ‘complete tree’ supplemental).
The authors make the interesting observation in Fig 1B that the most modified base is U. One possibility the authors could mention for the higher number of uracil-derived modified bases may be linked to the fact that C vs. U only have 1 ‘difference’ while G and A have 2. Therefore, any deamination step affecting the NH2 on the C4 of a C will produce a modified base that will look U-derived. Not linking these modifications to the C-tree would generate an over-representation of U-derived bases and an under-representation of C-derived bases, which is interestingly what the authors see in Fig 1B. In other words, there may be arrows linking some C bases to their U base equivalent, rather than treating these trees as independent. Some of these “trans”-steps are already known to occur in vivo, such as C to U, and m3C to m3U (Rubio et al, 2017, Nature).
It would also be important to clearly state both in the text and in Fig 1 which organisms are considered when establishing the tree: is it only eukaryotes or does it also include bacteria? Are viral RNA modifications also taken into account? On one hand, including more organisms may allow to extend the tree; on the other hand, some biosynthetic pathways may differ between species to form the same modified base – the model should therefore allow the tree to have converging branches. The m1A to m6A rearrangement could also be shown in Fig 2 (cf line 187-188). The text starting page 5 and onwards mentions bacteria & archaea so these must have been included, and in the text the species differences are already adequately mentioned (such as for o2yW). It could be important to stress somewhere in the text that we do not know whether modified bases will be located at the same sites, and will play the same biological roles, in all species.
For pseudouridine (line 346-347), the detection has been facilitated because the RT stops not from pseudouridine alone, but from treatment with CMC which acts as a RT block. The authors can cite (Carlile et al, 2014, Nature; and also ref 24) to illustrate this statement for example. In mRNA, this mark is targeted to a structural motif rather than a sequence motif; this could be indicated in Fig 4, as it would also make clear that all other modifications may likewise be affected by structure rather than sequence alone.
In the webtools section, the authors might want to add the useful tool HAMR (Ryvkin et al, 2013, RNA), which aims to detect diverse modifications types based on their RT-mutation profile (as discussed in ref 21 for example). This is especially useful considering that it seems that the webtools section (line 398-420 and Table 1) concentrates mostly on A-derived modifications at the moment.
In the introduction and conclusion, the authors mention that new assays will have to be developed to allow for a transcriptome-wide characterization of most marks. It may be worth to mention the promising development of epitranscriptomics assays in third-generation sequencing techniques, in particular nanopore-based platforms. In these approaches, modified bases will generate different current patterns when passing through the pore, and therefore have the potential to be read natively. A recent study managed to perform this for m6A (Liu et al, 2019, Nature Communications).
A bit of English editing is needed throughout the manuscript and in the abstract in order to make the text a bit clearer.
Minor typos:
p2, line 68: it is not exactly the RT that ‘erases’ a base; (it’s the information that is ‘erased’ not the base).
line 200: “3’UTRs”
Fig 1A: “unassigned”
Reviewer 3 Report
The manuscript entitled "A census and categorization method of epitranscriptomics marks" reviews the current knowledge on the epitranscriptome, while it also proposes a comprehensive categorization method of RNA modifications. This review is well written and focuses on a very pertinent scientific subject. RNA modifications are becoming central in biology, as they constitute a post-transcriptional regulatory layer of gene expression. The categorization the authors propose is extremely useful in particular for researchers that are not so familiar with RNA modifications. In general terms, this manuscript is suitable for publication, however, I would like to leave some suggestions to further improve the manuscript:
1) The figure resolution should be improved;
2) on line 166 "....as the least prevalent epitranscriptomes in tRNAs..." should be substituted by "...as the least prevalent epitranscriptomic changes in tRNAs..."
3) on line 308 "...are occurs also in sncRNAs and lncRNAs..." should be substituted by "..occurs also in sncRNAs and lncRNAs.."
4) Sections concerning Adenine, Cytosine and Guanine-derived modifications are very comprehensive, however, Uracil-derived modification section is not. Considering that, at least for tRNAs, U is highly modified by a panoply of enzymes and in different positions, and is also highly correlated to human disease, this could be explored in this section.
5) The Epitranscriptomics role in physiology and disease also has room to improve. Although this is not the focus of this review, recent data correlates both RNA, rRNA and tRNA epitranscriptomic changes to a panoply of diseases that are only superficially mentioned in this section. Moreover, neurological disorders such as familial dysautonomia, ALS, among others are directly linked to modification deregulation and are not mentioned in this section. A more concrete paragraph on this section mentioning the findings correlating epitranscriptomic changes to diseases would largely improve the manuscript.
6) section numbering needs revision, as most sections are numbered as "2", namely "a categorization method for RNA modifications"; "adenine-derived modifications", "cytosine-derived modifications", "guanine-derived modifications" and "uracil-derived modifications";
Reviewer 4 Report
The work by Mathlin and Colombo summarizes current knowledge on RNA modifications and present a novel RNA modification categorization scheme that can be further modified in the future as more RNA modifications are uncovered. The Review article is interesting and valuable, especially for the ease of access to the information the authors present both in their categorization method for RNA modifications, as well as the selected databases and webtool currently available for exploring RNA modifications. I have some comments that I believe will strengthen the manuscript and that the authors should address.
The authors correctly emphasize that our current knowledge on RNA modifications rely on HTS technologies that require reverse-transcription (line 68/69, page 2). In this sense, it would be good to mention that Nanopore technology is emerging and show promising results to avoid reverse-transcription caveats (see for example Xu, L and Seki, M J Hum Genet 2020). Another issue that is worth mentioning is the fact that RNA modifications can bias both transcript and modification quantification when HTS technologies are used (see for example Cozen A et al Nat Methods 2015 and Zheng G et al. Nat Methods 2015). The authors should expand more on the queosine modification. Explain why Figure 1 shows 9 different queosine modification. My understanding is that there are only three derivatives of Queosine (Galactosyl-, Mannosyl- and Glutamyl-) in addition to Queosine itself; therefore that number should be 4 instead of 9. Please include reference to the appropriate literature if more Q derivatives exist. Furthermore, I believe that Queosine modification occurs only at the wobble position of tRNAs with GUN anticodons and, in the author’s classification should be incorporated as a G-derivative. Regarding the “lack of information of their biosynthesis” as the authors argue; in my opinion the biosynthesis of Q is pretty well characterized and should be discussed. Even though incorporation of Q in tRNAs depend on a transglycosylation reaction that replaces the G residue instead of modifying it, I believe it should be considered a G-derivative of second stage according to the authors’ classification of RNA modifications with the 3 additional Q-derivatives as 3rd stage modifications. See for example Fergus C et al Nutrients 2015 and also the primary research cited in that work. Also please explain in the text which are the 17 phosphate modifications that Figure 1 shows, and please verify that indeed no enough information on their biosynthesis route exist. Page 3, line 114. Does a given enzyme catalise only one stage of only one type of modification? If so, can this classification be also used to know how many different enzymes are required for any given RNA modification? In the m1A section, the authors should also mention ALKBH1 as a demethylase (see Liu F et al Cell 2016). Figure 1 as this Reviewer has seen it is in black and white format and therefore the bar plot has confusing coloring (e.g. stage 10th, 6th and 2nd look alike). Please amend this Figure so that is can be interpreted more easily. The section on A-to-I editing needs thorough modification: Differentiate between the enzymes that catalyse A-to-I editing in tRNAs and in other RNAs. ADARs (RNAs) and ADAT1 (m1I37) have an adenosine deaminase domain, while ADAT2 and ADAT3 (I34) have a cytidine deaminase domain, and are therefore very different enzymes. Please see Torres AG FEBS Lett 2014. Page 6, line 223. Targeting of double-stranded RNAs is only for ADARs not ADATs. This needs to be modified, it is also one of the main differences between ADARs and ADAT1 (that both have adenosine deaminase domains but ADAT1 lacks double-stranded RNA binding properties). Inosine at position 37 is found only on eukaryotic tRNA Ala and is always modified to m1I. As far as I know Im37 does not exist, if it does please cite the appropriate literature. Reference 49 (Torres AG Nucleic Acids Res 2015) does not address in depth inosine at position 37. Perhaps the authors wanted to refer to the Review article Torres AG et al FEBS Lett 2014. Reference 49 shows that I34 and m1I37 can be detected by HTS and should be mentioned in the text (along reference 50 that indicates that m1I37 can be detected by HTS). Please give more detail on the sentence “In contrast, the function of the m1I modification is unknown, yet ADAT1 knock-out yeasts are shown to be viable, indicating that role of m1I modification is not as essential for survival as inosine”. Are the authors comparing the essentiality of ADAT1 against ADAT2/ADAT3? That is m1I37 vs I34? If so, please indicate this properly and cite the appropriate literature… I did not see the yeast KO experiment in the cited article reference 49 (perhaps again the authors wanted to cite the inosine Review mentioned above, Torres AG et al FEBS Lett 2014?). Page 6, line 229. The authors can be more specific. I34 is found in all tRNAs genetically encoded with A34 in eukaryotes (8 substrates) and in 2 substrates in Bacteria (Arg, Leu) (See Rafels-Ybern et al Mol Biol Evol 2019). Page 5, line 176. The authors mention the formation of m1I57 in archaea in this section. It should be mentioned in the A-to-I editing section. Moreover, for specific cases like this one I encourage the authors to cite the primary work (e.g. Grosjean H. et al Nucleic Acids Res. 1995). M1I57 can be further methylated into m1Im57 in thermophilic organisms (see Edmonds CG J Bacteriol 1991). This modification should also be mentioned in the text, and added to the corresponding classification. Reference 51 (Cattenoz et al RNA 2013) refers to A-to-I editing in mRNAs. A-to-I editing on tRNAs is also important in nervous system function. See for example Alazami AM et al J Med Genet 2013 and other articles on ADAT3-related intellectual disability.
9. In the cytosine methylation section the authors should also cite the work from Michaela Frye’s lab (e.g. Blanco, S. et al Nature 2016, Blanco, S. EMBO J 2014).
10. Figure 2 is in my opinion the most important Figure and it is too small. Please make sure that all modifications can be easily read. In addition, I think it would be good to show the different biosynthetic routes leading to the same modified nucleoside. For example, m1I appears twice in Figure 2B because it can result from A-to-I editing followed by methylation, or by A-methylation followed by mA-to-mI editing. It will simplify the figure if in these cases, the second stage modification is retain and then arrows departing from those modifications leads to the same 3rd level modification indicated only once.
11. Legend on Figure 3 should not have the title “RNA modifications on eukaryotic tRNA” because it also shows bacterial and archaeal modifications. The Figure as seen in the file this Reviewer has evaluated, there are only few modifications underlined (meaning “known to alter codon recognition”), authors should either remove that information or underline all modifications known to alter codon recognition (e.g. all modifications at position 34 and many modifications at position 37 at least). In addition, Figure 3a should also include m1I57 and m1Im57 as discussed above. Also, if I am not mistaken, Im37 does not exist and should therefore be removed from this Figure.
12. Epitranscriptomics role in physiology and disease. This section is very poor but I understand the difficulty in compiling thorough information on this matter that may be already out of the scope of the manuscript. In any case, the authors are in this section focusing on modifications mainly on mRNAs. They should also at least mention that tRNA modifications (beyond MELAS and MERRFS) are extremely important for physiology and disease (see for example Torres AG et al Trends Mol Med 2014) and also modifications in other RNAs such as piRNAs, miRNAs and tsRNAs (see for example Zhang X et al Trends Mol Med 2016).
13. What does “Reference” in Table 1 mean? It would be best if the article reporting the database/webtool is cited instead of this long numbering code.
Additional comments:
14. I could not access Supplemental Table 1, and was therefore not revised by this Reviewer.
15. Although not dramatically necessary, I think it would be helpful for the non-specialized reader to have an additional figure showing the chemical structure of the main types of RNA modifications (e.g. 2’-0-methylaiton, thiol modifications, inosine, etc.)
16. Typos: page 3 line 11: should read “the third-stage has one modification added on top of the second state, and so on…”. Page 4, line 149 “Quanosine-derivatives” should read “Queosine-derivatives”. Page 4, line 150 should read “…are not allocated in this scheme due to the current lack of information…”. Page 5, line 200 “3’-UTSs” should read “3’-UTRs”. Page 8, line 307 should read “In addition to tRNAs, m2G modifications occur also in snRNAs…”. Page 9, line 351, I think “conversed” should read “conserved”.